# Anemia-Associated Platelets and Plasma Prothrombin Time Increase in Patients with Adenomyosis

**DOI:** 10.3390/jcm11154382

**Published:** 2022-07-28

**Authors:** Qiao Lin, Tiantian Li, ShaoJie Ding, Qin Yu, Xinmei Zhang

**Affiliations:** 1Women’s Hospital, School of Medicine, Zhejiang University, Hangzhou 310006, China; y214180202@zju.edu.cn (Q.L.); 11718248@zju.edu.cn (T.L.); dsjd12345@zju.edu.cn (S.D.); qinyu2019@zju.edu.cn (Q.Y.); 2Taizhou Cancer Hospital, Taizhou 317502, China

**Keywords:** adenomyosis, coagulation parameters, anemia, platelets

## Abstract

Patients with adenomyosis are hypercoagulable and often accompanied by anemia, but the specific changes in anemia-related coagulation parameters are still unclear. This study investigated the changes in and influencing factors of coagulation parameters related to anemia in patients with adenomyosis (AM). The coagulation parameters, including platelet count (PC), plasma prothrombin time (PT), activated partial prothrombin time (APTT), thrombin time (TT) and fibrinogen (FB), and hemoglobin (Hb), were measured in patients with adenomyosis (229 cases in AM group), uterine leiomyoma (265 cases in LM group), and undergoing tubal anastomosis (142 cases in the control group). The age of the control group was younger than that of the AM group and the LM group. Compared with the AM and LM groups, the uterus size of the control group was smaller; the AM group was larger than the LM group. The Hb concentration of the AM group was lower than that of the LM and control groups. Compared with the LM and control groups, PC increased and TT shortened in the AM group. APTT in the AM group was shorter than in the control group, and PT was longer than in the LM group. After adjustment using multiple logistic regression analysis, adenomyosis was correlated with Hb concentration (or = 0.971, 95% CI 0.954–0.988, *p* < 0.001), PC (or = 1.006, 95% CI 1.002–1.011, *p* = 0.004), PT (or = 3.878, 95% CI 2.347–6.409, *p* < 0.001), age (or = 1.062, 95% CI 1.013–1.114, *p* = 0.013), and uterine size (or = 1.103, 95% CI 1.011–1.203, *p* = 0.028). Correlation analysis showed that PC (r = −0.309) and PT (r = −0.252) were negatively correlated with anemia. The increase in Hb-related PC and PT in patients with adenomyosis indicates that the timely and early detection of coagulation parameters is needed for patients with severe anemia, older age, and larger uterine volume.

## 1. Introduction

Adenomyosis is a common benign gynecological disease in which endometrial tissue (glands and stroma) grows within the myometrium of the uterus, resulting in hypertrophy of the myometrial smooth muscle [1]. The main manifestations of adenomyosis are menorrhagia, dysmenorrhea, and infertility [1,2]. Currently, the tissue injury and repair (TIAR) mechanism is the most widely accepted pathological explanation, although the etiology and pathogenesis of adenomyosis remain unclear [1,2,3,4,5]. Recently, the hypercoagulability state, including changes in thrombin spectrum and platelet count and volume, has been observed in patients with adenomyosis, which may further elucidate the pathogenesis, diagnosis, and management of the disease [6,7,8,9]. However, it is still unclear whether the hypercoagulability state is the cause or effect of adenomyosis.

It has been found that platelets appear in adenomyosis lesions but not in healthy endometrial tissue, suggesting that activated platelets and the associated aggregation play a role in the pathogenesis of adenomyosis [10]. Further studies have shown that platelets can impair the reactivity and function of natural killer cells [11], increase the production of estrogen [12], and induce the synthesis of inflammatory factors and angiogenic factors by mediating the epithelial to mesenchymal transition and fibroblast to myofibroblast transition via the transforming growth factor-beta-1 signaling pathway [10,11,12,13,14], leading to the development and progression of endometriosis. As such, anti-platelet therapy is considered a promising method for the clinical treatment of adenomyosis [15]. However, a recent study did not demonstrate the presence of platelets in adenomyotic lesions [4]. Evidently, whether platelets are involved in the pathogenesis of adenomyosis still requires further investigation.

In patients with endometriosis, activated partial prothrombin time (APTT), prothrombin time (PT), and thrombin time (TT) were significantly shortened, but fibrinogen (FB) significantly increased, indicating that women with endometriosis are in a hypercoagulable state [16]. The levels of fibrinogen factor XIII and antithrombin-III are significantly higher in women with adenomyosis than those in women with endometriosis [17]. Hence, the coagulation parameters of adenomyosis, such as endometriosis, including platelet count (PC), APTT, PT, TT, and FB, are also helpful for its clinical diagnosis, although the exact mechanism of the hypercoagulable state of adenomyosis remains unclear [6,7,8,9]. Recently, a recent systematic review by Ottolina et al. did not find any significant difference in PC between patients with and without endometriosis, regardless of the disease stage and type [18]. In comparison to women without endometriosis, women with ovarian endometriosis have lower PC values but higher mean platelet volume (MPV) [19]. A study by Bodur et al. found that MPV, rather than PC, increased in women with adenomyosis, suggesting that MPV can be used as a diagnostic marker of this disease [20]. However, a case–control study by Coskun et al. showed that PC and MPV values were not related to the presence of endometriosis or adenomyosis [21]. Apparently, the measure of platelets as an auxiliary diagnosis of adenomyosis and endometriosis still needs further investigation.

Since adenomyosis and endometriosis are chronic inflammatory diseases, coagulation parameters of adenomyosis may be related to inflammatory parameters [22]. Inflammation can trigger the coagulation system by reducing the activity of natural anticoagulation mechanisms and impairing the fibrinolytic system. Therefore, inflammatory cytokines are the main mediators involved in coagulation activation [23]. In addition to inflammation, anemia can impair the quality and activity of red blood cells (RBCs), thus affecting the changes in coagulation parameters involved in the coagulation mechanism of the disease [24,25,26]. It has been shown that iron deficiency anemia (IDA) and the use of oral contraceptives can induce transferrin overexpression through hypoxia and estrogen response elements, respectively, thus promoting hypercoagulability [27]. Moreover, the interaction of platelets and RBCs can induce procoagulant activity through the FasL/FasR signaling pathway [28]. Clinically, patients with adenomyosis present menorrhagia during menstruation, leading to severe anemia and secondary complications such as disseminated intravascular coagulation, acute renal failure, and cerebral infarction; this may explain the role of anemia in the activation of the coagulation mechanism [29,30,31]. Considering that adenomysis and uterine leiomyoma are estrogen-dependent benign diseases often comorbid with menorrhagic anemia [32], in the present study, we conducted a cross-sectional study to determine the changes in coagulation parameters, including PC, PT, APTT, TT, and FB, in women with adenomyosis as compared to women with uterine leiomyoma and healthy women who underwent tubal anastomosis. We then analyzed the relevant factors affecting these changes.

## 2. Materials and Methods

### 2.1. Patients

Between January 2015 and December 2017, a total of 997 patients who underwent laparoscopic/laparotomic surgery for adenomyosis (*n* = 395), uterine leiomyoma (*n* = 437), and tubal anastomosis (*n* = 165) at the Women’s Hospital at the Zhejiang University School of Medicine were recruited in this study. The general information and clinical data of all patients, including age, body mass index (BMI), gravidity, incidence of abortion, menorrhagia, dysmenorrhea, pelvic adhesion, hemoglobin levels, uterine size, and anemia, were retrospectively obtained and recorded from the original electronic medical record (EMR) of hospitalized patients. The study was approved by the Human Ethics Committee of the Women’s Hospital at the Zhejiang University School of Medicine (No. 20170174).

Patients with uterine leiomyoma and adenomyosis were diagnosed by ultrasound and nuclear magnetic resonance imaging (MRI) preoperatively and confirmed by surgery and postoperative histopathology. For patients who underwent tubal anastomosis, routine ultrasound examination was performed before operation, which was postoperatively confirmed by surgical findings and histopathology. Of the 395 patients with adenomyosis, 166 patients were excluded because of endometriosis (*n* = 117), preoperative hormone therapy (*n* = 48), and postmenopause (*n* = 1). Similarly, 172 of 437 patients with uterine leiomyoma were excluded due to endometriosis (*n* = 113), adenomyosis (*n* = 49), and postmenopause (*n* = 10). In addition, 23 of the 165 patients who underwent tubal anastomosis were excluded, including endometriosis (17 cases) and adenomyosis (6 cases). As a result, the remaining 636 patients consisting of 229 cases of adenomyosis (AM group), 265 cases of uterine leiomyoma (LM group), and 142 cases of tubal anastomosis (control group) were included in this study (Figure 1). All patients had no previous history of cancer, hypertension, diabetes, coagulation dysfunction, autoimmune disease, and kidney disease, and did not use anticoagulants or iron supplement therapy 6 months before blood sample collection.

### 2.2. Blood Assays

All patients were hospitalized during the non-menstrual period and underwent surgery. Peripheral blood samples were collected from the median elbow vein before surgery. Hemoglobin concentration (Hb, range: 110–150 g/L) and platelet count (PC, range: 100–300 × 10^9^/L) were measured by an automatic classification analyzer (Beckman, Coulter LH750). Coagulation parameters, including plasma prothrombin time (PT, range: 11.5–14.3 s), activated partial prothrombin time (APTT, range: 28–40 s), thrombin time (TT, range: 13.5–18.5 s), and fibrinogen (FB, range: 2–4 g/L), were measured by an automatic blood-coagulation analyzer (STAGO, Evolution ISTA-R-IV, France, Germany).

### 2.3. Determination of Uterine Size

The size of the uterus was measured by ultrasonography (uterine volume = A × B × C × 0.5233 (where A, B, and C are the uterine length, width, and thickness, respectively)).

### 2.4. Statistical Analysis

The SPSS 25.0 package was used for statistical analysis and *p* < 0.05 was considered to be statistically significant. The Shapiro–Wilk test was used to determine whether continuous variables had normal distribution. The continuous variables in the study were skewed distribution variables and were presented as the median with an interquartile range [M (P25–75)]. The intragroup differences were compared by using Kruskal–Wallis test. The Mann–Whitney U and χ^2^ tests were used to compare the medians and frequencies between the AM group, LM group, and the control group. Moreover, χ^2^ tests were used for univariate analysis. Logistic regression was used for multivariate analysis, in which the covariates included were those found to be statistically significant in the univariate analysis. In this study, no continuous variables were categorized except for the hemoglobin level. A Spearman’s rank correlation coefficient was conducted to determine the correlations between coagulation parameters and hemoglobin level.

This study used the formula for mismatched case-control studies (α = 0.05, β = 0.1) to calculate the sample size and determine the sample size with PT as the primary outcome. In our small sample test, according to the cut-off of 13.0 s, PT, for the diagnosis of adenomyosis, p1 and p2 were 0.65 and 0.3, respectively, the required sample size was at least 61 women per group.

## 3. Results

### 3.1. Patients’ Characteristics

Of the 229 patients with adenomyosis, 206 (90.0%) were diffuse and 23 (10.0%) were focal. Among 265 patients with uterine leiomyoma, 15 (5.7%) cases were subserosal, 139 (52.5%) cases were intramural, 13 (4.9%) cases were submucous, 26 (9.8%) cases were subserosal and intramural, 23 (8.7%) cases were subserosal and intramural, and 49 (18.5%) cases were subserosal, submucous, and intramural. There was no significant difference in BMI among the three groups (*p* = 0.209). The age of the control group was significantly younger than that of the AM group (*p* < 0.001) and the LM group (*p* < 0.001). Compared with the LM and control groups (*p* all < 0.001), more patients in the AM group suffered from dysmenorrhea and menorrhagia. Moreover, lower hemoglobin concentration and anemia were more common in patients with adenomyosis compared to the LM group (*p* < 0.001) and the control group (*p* < 0.001). The pelvic adhesion rate of the control group was significantly higher than that of the AM group (*p* < 0.001) and the LM group (*p* < 0.001), but there was no significant difference between the LM and AM groups (*p* > 0.05). Compared with the control group, the uterine volumes of the AM and LM groups were larger (*p* < 0.001). In addition, the size of uterus in the LM group was smaller than that in the AM group (*p* < 0.001, Table 1).

### 3.2. Comparisons of Coagulation Parameters between Groups

The number of PC in the AM group was significantly higher than that in the LM group (*p* < 0.001) and the control group (*p* < 0.001). The median time of TT was significantly shorter in the AM group than that in the LM group (*p* < 0.001) and the control group (*p* < 0.001). Moreover, the median time of PT was significantly longer in the AM group than that in the LM group (*p* < 0.001), while APTT in the AM group was shorter than in the control group (*p* < 0.001, Table 2, Figure 2).

### 3.3. Correlations of Coagulation Parameters and Adenomyosis

In order to further determine whether changes in coagulation parameters are caused by adenomyosis, we conducted a multiple logistic regression model (LM group and control group were coded as 0; AM group was coded as 1) because there were significant differences in the characteristics of the three groups. The logistic regression analysis showed that adenomyosis significantly correlated with Hb concentration (or = 0.971, 95% CI 0.954–0.988, *p* < 0.001), PC (or = 1.006, 95% CI 1.002–1.011, *p* = 0.004), and PT (or = 3.878, 95% CI 2.347–6.409, *p* < 0.001) but did not correlate with APTT, TT, or FB (*p* > 0.05). In addition, age (or = 1.062, 95% CI 1.013–1.114, *p* = 0.013), abortion (or = 1.913, 95% CI 1.189–3.078, *p* = 0.008), dysmenorrhea (or = 30.203, 95% CI 16.937–53.859, *p* < 0.001), and uterine size (or = 1.103, 95% CI 1.011–1.203, *p* = 0.028) were high-risk factors for adenomyosis (see Table 3).

### 3.4. Correlation of PC and PT with Anemia in Patients with Adenomyosis

In order to analyze the correlation of PC and PT with anemia in women with adenomyosis, Spearman’s analysis was used to analyze potential correlations. Figure 3 showed that both PC (r = −0.309, *p* < 0.001) and PT (r = −0.252, *p* < 0.001) were negatively correlated with hemoglobin concentration. 

### 3.5. Predictive Value of PC and PT in the Diagnosis of Adenomyosis

Based on the correlations of PC and PT with adenomyosis, we used PC and PT as predictive values in the diagnosis of adenomyosis. The AUC of PC was 0.685 (95% CI 0.641–0.728) with a sensitivity of 53.7%, a specificity of 75.4%, and a cut-off value of 275.5 × 10^9^/L. The AUC of PT was 0.673 (95% CI 0.631–0.716) with a sensitivity of 59.8%, a specificity of 67.3%, and a cut-off value of 13.0 s (Figure 4).

## 4. Discussion

Our results showed that PC increased in women with adenomyosis compared with healthy women, which is consistent with the results of Zhang et al. [7] for women in the non-menstrual period. Moreover, the PC value of patients with adenomyosis was higher than that of patients with uterine leiomyoma, although both are estrogen-dependent benign diseases. Multiple logistic regression analysis was used to further confirm that adenomyosis was associated with PC rather than uterine leiomyoma. Our results also showed that the platelet count was negatively correlated with hemoglobin concentration, indicating that increased PC in women with adenomyosis is related to anemia, corroborating the results of Zhang et al. [9]. Although previous studies by Bodur et al. [20] and Coskun et al. [21] did not find the correlation between the increase in PC value and patients with adenomyosis, their studies either involved patients with adenomyosis without anemia or did not provide anemia data. In this study, we found that the incidence of menorrhagia and anemia were higher in women with adenomyosis compared to women with uterine leiomyoma and women undergoing tubal anastomosis. In fact, anemia caused by menorrhagia in patients with adenomyosis belongs to iron deficiency anemia (IDA). Iron deficiency anemia can reactively cause platelet increase [33]. Since an ultrasound shows that the severity of adenomyosis is positively correlated with menstrual blood loss to a significant extent [34], this finding, together with our results, suggests that the number of platelets may be related to the severity of adenomyosis.

In addition to reactive PC increases, iron deficiency (ID) can promote thrombotic tendency [35]. It has been found that the ID, IDA, and oral contraceptives (OCs) induce transferrin overexpression through the PI3K/AKT signaling pathway, resulting in a significant reduction in APTT, PT, and plasma recalcification time (PRT) and a significant increase in PC and platelet-based thrombin generation; anti-transferrin antibody treatment significantly reverses the reduction in APTT, PT, and PRT, and the elevation of PC and platelet-based thrombin generation [27]. In fact, iron-supplement therapy for anemia also affects the changes of coagulation parameters [36]. In this study, we found that PT was significantly longer in women with adenomyosis than that in women with uterine leiomyoma and women undergoing tubal anastomosis. Further, PT was negatively correlated with anemia in women with adenomyosis. Although anemia caused by menorrhagia in patients with adenomyosis and uterine leiomyoma is widely considered to be iron deficiency anemia, iron was not supplemented 6 months before blood samples were collected in this study, and concentrations of iron and transferrin were not detected. Recently, one study found that PT significantly shortened only during the menstrual bleeding period compared with the control group [7]. Another study did not find any significant differences in PT between patients with adenomyosis and patients with uterine leiomyoma but found that TT was significantly shorter in adenomyosis with anemia than that in uterine leiomyoma with anemia [9]. Since normal red blood cells (RBCs) can promote thrombus formation and enhance thrombus stability [24], impaired RBCs caused by anemia may lead to the delay of coagulation cascade [25]. Moreover, the interaction of platelet-RBCs can induce procoagulant activity through the FasL/FasR signaling pathway [28]. Thus, anemia can affect parameters involved in the coagulation mechanism of adenomyosis [24,25,26].

In addition to anemia, the changes of coagulation parameters in women with adenomyosis are also affected by many factors. Our study found that age, abortion, dysmenorrhea, and uterine size were high-risk factors for adenomyosis. It has been demonstrated that adenomyosis patients with a uterus volume ≥ 100 cubic centimeters are at risk of having an activated coagulation system [6]. A recent study by Piccioni et al. showed that the frequency of menorrhagia in adenomyosis patients >35 years old is almost twice that in adenomyosis patients ≤35 years old [37]. It is suggested that patients with adenomyosis tend to be younger in the early stage of the disease, while those in the advanced stage of the disease tend to be older, indicating that age may also reflect the progress of the disease. Adenomyosis often coexists with endometriosis [38,39], and endometriotic stroma cells can secrete thrombin and thromboxane A2, inducing platelet activation and aggregation [40]. Hence, endometriosis may affect the results if adenomyosis is comorbid with endometriosis. Therefore, patients with endometriosis were excluded from the study.

Adenomyosis is a chronic inflammatory disease wherein inflammation can impair the fibrinolytic system and reduce the activity of natural anticoagulation mechanisms, thus activating the coagulation system [23]. Additionally, OCs are used to treat estrogen-dependent diseases, including adenomyosis, endometriosis, and uterine leiomyoma, which can induce hypercoagulability states [27]. A series of clinical case reports of patients with adenomyosis and uterine leiomyoma complicated by severe menorrhagic anemia leading to diffuse intravascular coagulation, acute renal failure, cerebral infarcts, and acute ischemic stroke during menstruation strongly support the hypothesis that anemia induces hypercoagulability states [29,30,31,41,42,43]. It is suggested that prolonged PT and increased PC may be the consequence of adenomyosis. Similarly, there are many factors that can increase platelet count. Ectopic endometrium-associated platelets in endometriosis and adenomyosis may be related to the increase in platelets [10,13]. Increased platelets can promote inflammation and angiogenesis, which is not only related to the pathogenesis of adenomyosis and endometriosis but also related to the occurrence of malignant tumors [10,13,44]. Nevertheless, our results found that the increase in PC and PT were negatively correlated with anemia, indicating that the more severe adenomyosis, the higher the PC and the longer the PT. The early detection of coagulation parameters may be useful for the diagnosis, prevention, and treatment of adenomyosis.

However, due to the retrospective nature of the study, some data that may affect coagulation parameters, such as iron and transferrin, CA125 and CA199, adenomyosis subtype, uterine leiomyoma subtype, and menstrual coagulation parameters, could not be completely obtained. More importantly, we only collected data from patients with adenomyosis and uterine leiomyoma who underwent laparoscopic and/or laparotomic surgery; we did not collect data from patients with submucosal leiomyoma that may cause severe anemia requiring hysteroscopic surgery. As a result, the anemia rate of patients with uterine leiomyoma was not high. In this manner, we could not perform a comparison between uterine leiomyoma with anemia and adenomyosis with anemia, which may affect the results of this study. Therefore, it is necessary to conduct a prospective study of large samples, including factors that may affect coagulation parameters, to further verify the results of this study.

In summary, our results showed that the PC and PT increased in women with adenomyosis and negatively correlated with anemia, indicating that the timely and early detection of coagulation parameters and strict monitoring of patients with adenomyosis who have severe anemia or a long course of disease are needed. However, many factors, including anemia, can affect the coagulation parameters. Therefore, prospective large-sample multicenter studies are needed.

## Figures and Tables

**Figure 1 jcm-11-04382-f001:**
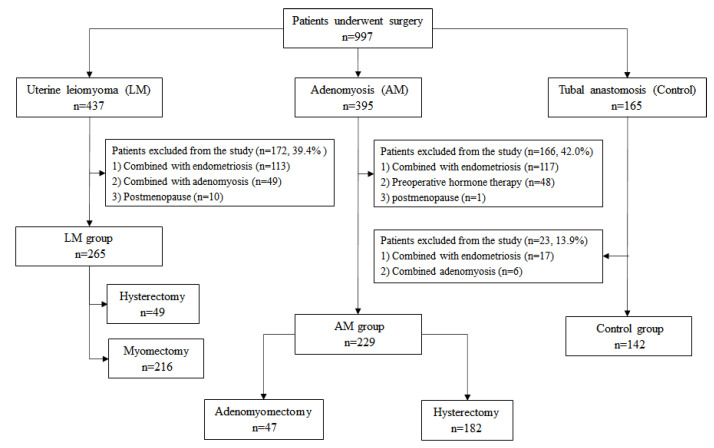
Study flow diagram.

**Figure 2 jcm-11-04382-f002:**
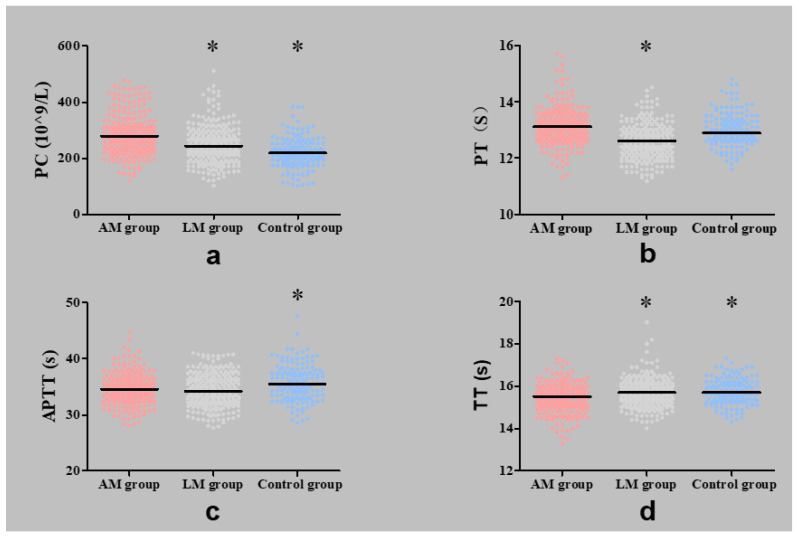
Comparisons of coagulation parameters between groups (median and range). (**a**) Comparison of platelet count (PC) between groups; (**b**) Comparison of plasm a prothrombin time (PT) between groups; (**c**) Comparison of activated partial prothrombin time (APTT) between groups; (**d**) Comparison of thrombin time (TT) between groups. *, *p* < 0.025 (compared with the AM group).

**Figure 3 jcm-11-04382-f003:**
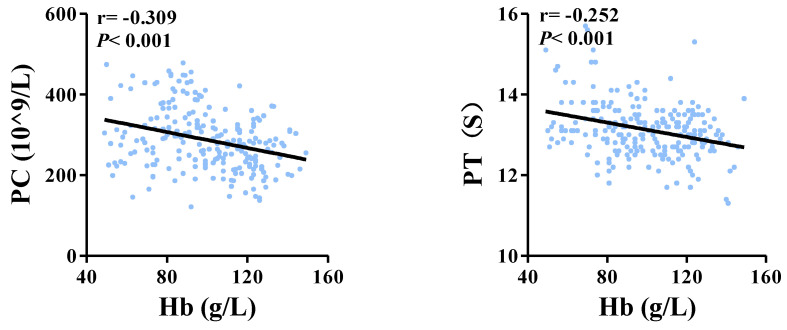
Correlations of PC and PT with hemoglobin PC, platelet count; PT, plasma prothrombin time.

**Figure 4 jcm-11-04382-f004:**
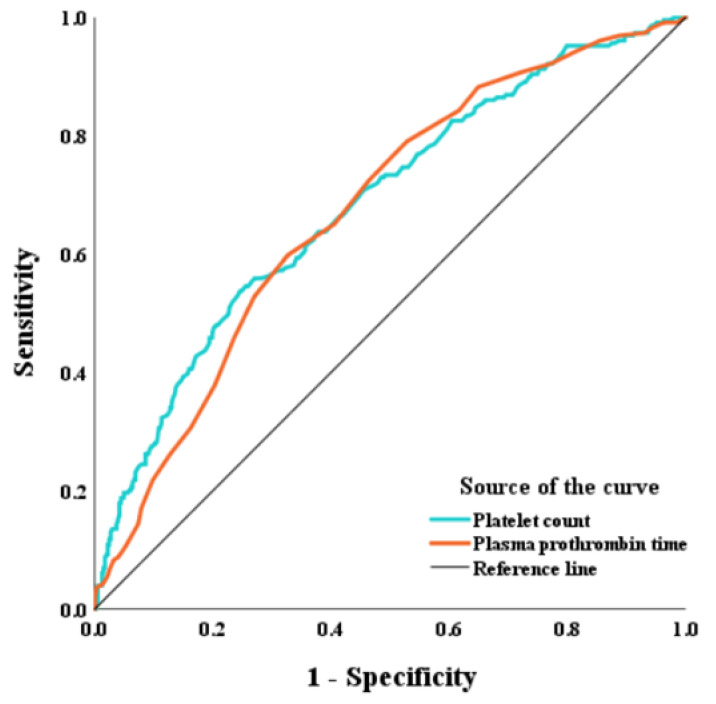
Receiver-operating characteristic curves of PC and PT for the diagnosis of adenomyosis.

**Table 1 jcm-11-04382-t001:** Patients’ characteristics.

Variables	AM Group (*n* = 229)	LM Group (*n* = 265)	Control Group (*n* = 142)	*p*-Value
Age (years)	44 (40–48)	44 (39–46)	36 (32–39) *	<0.001
BMI (kg/m^2^)	23.0 (21.1–25.0)	22.6 (21.1–24.6)	22.7 (20.8–24.5)	0.209
Gravidity	3 (2–4)	3 (2–4) *	3 (2–4) *	<0.001
Abortion	2 (1–3)	1(1–2) *	1 (0–2) *	<0.001
Menorrhagia (*n*, %)	109 (47.6)	83 (31.3) *	7 (4.9) *	<0.001
Anemia (*n*, %)	141 (61.6)	60 (22.6) *	11 (7.7) *	<0.001
Hb (g/L)	99.0 (81.0–119.0)	124.0 (110.5–133.0) *	126.0 (118.8–134.0) *	<0.001
Dysmenorrhea (*n*, %)	190 (83.0)	41 (15.5) *	13 (9.2) *	<0.001
Pelvic adhesion (*n*, %)	70 (30.6)	83 (31.4)	72 (50.7) *	<0.001
Uterine size (cm^3^)	245.6 (174.3–435.2)	204.5 (173.2–244.9) *	48.5 (45.7–51.2) *	<0.001

Notes: Data were shown as M (P25–75) and *n* (%) as appropriate; *, *p* < 0.025 (compared with the AM group); AM group, patients with adenomyosis; LM group, patients with uterine leiomyoma; Control group, patients undergoing tubal anastomosis; BMI, body mass index; Hb, hemoglobin.

**Table 2 jcm-11-04382-t002:** Comparisons of coagulation parameters between groups.

Parameters	AM Group (*n* = 229)	LM Group (*n* = 265)	Control Group (*n* = 142)	*p*-Value
PC (10^9^/L)	279.0 (230.0–327.0)	244.0 (208.5–284.5)*	218.5 (182.5–254.3)*	<0.001
PT (s)	13.1 (12.7–13.5)	12.6 (12.2–12.9) *	12.9 (12.6–13.4)	<0.001
APTT (s)	34.6 (32.8–36.3)	34.2 (32.2–36.6)	35.6 (33.4–37.8) *	<0.001
TT (s)	15.5 (15.0–15.9)	15.7 (15.3–16.1) *	15.7 (15.3–16.2) *	<0.001
FB (g/L)	2.9 (2.6–3.3)	3.0 (2.7–3.3)	2.8 (2.5–3.1)	0.066

Notes: Data were expressed as M (P25–75); *, *p* < 0.025 (compared with AM group); AM group, patients with adenomyosis; LM group, patients with uterine leiomyoma; Control group, patients undergoing tubal anastomosis; PC, platelet count; PT, plasma prothrombin time; APTT, activated partial prothrombin time; TT, thrombin time; FB, fibrinogen.

**Table 3 jcm-11-04382-t003:** Correlations of coagulation parameters and adenomyosis.

Covariates	Estimates	SE	Odds Ratio (95% CI)	*p*-Value
Hb	−0.030	0.009	0.971 (0.954–0.988)	0.001
PC	0.006	0.002	1.006 (1.002–1.011)	0.004
PT	1.355	0.256	3.878 (2.347–6.409)	<0.001
APTT	−0.024	0.050	0.976 (0.884–1.077)	0.628
TT	0.136	0.239	1.145 (0.717–1.830)	0.571
FB	0.351	0.307	1.420 (0.778–2.592)	0.254
Age	0.060	0.024	1.062 (1.013–1.114)	0.013
Gravidity	−0.304	0.220	0.738 (0.479–1.137)	0.168
Abortion	0.649	0.243	1.913 (1.189–3.078)	0.008
Menorrhagia	0.030	0.326	1.030 (0.544–1.951)	0.927
Dysmenorrhea	3.408	0.295	30.203 (16.937–53.859)	<0.001
Pelvic adhesion	0.286	0.301	1.331 (0.738–2.400)	0.342
Uterine size	0.098	0.044	1.103 (1.011–1.203)	0.028

Notes: Hb, hemoglobin; PC, platelet count; PT, plasma prothrombin time; APTT, activated partial prothrombin time; TT, thrombin time; FB, fibrinogen; SE, standard error.

## Data Availability

Our hospital electronic medical record.

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
