# Peer review of "Anemia-Associated Platelets and Plasma Prothrombin Time Increase in Patients with Adenomyosis"

_jcm, 2022, doi:10.3390/jcm11154382_

Round 1

Reviewer 1 Report

This manuscript could be a useful article examining the changes of coagulation parameters and associated factors in women with adenomyosis. However, there is no description of why the changes in coagulation parameters and associated factors in women with uterine adenomyosis should be identified. There is mention in previous papers that they have not yet been clarified. but the manuscript does not indicate the benefit to patients with adenomyosis after it is clarified. It is not important to clarify because it is not known, but what can be done for adenomyosis if it is clarified. A quicker diagnosis? A safe treatment plan?  In other words, it does not describe the medical problems (research questions) caused by the lack of identification of changes in coagulation parameters and associated factors.

       It also compares the leiomyoma group as well as the control group, but there is no mention of the need for that comparison. The comparison may have been made because of the similarity to adenomyosis in that it is a benign disease with an enlarged uterus, heavy menstrual bleeding (menorrhagia) and anemia. This may be to indicate whether the cause of the change in coagulation parameters is adenomyosis or heavy menstrual bleeding and/or anemia. However, it is unclear whether the myoma group is worthy of comparison since there is no mention of uterine size, myoma type, or presence of heavy menstrual bleeding. Conversely, the myometrial group in this study had less anemia and significantly higher hemoglobin levels, suggesting that the myometrial group in this study is inappropriate for comparison. Cases of myoma with a uterus approximately the same size as adenomyosis and approximately the same frequency of excessive menstrual periods should be selected (e.g., only intramuscular myomas of the same size as adenomyosis should be selected). 

The following is a list of points that should be corrected.

1.    Introduction

1)     The lack of clarification of changes in coagulation parameters and related factors in adenomyosis is described by presenting many references, but what would be improved or what would be useful if this were clarified should be described.

2)     The reason for comparing adenomyosis to the control group has been clarified, but the significance of comparing the uterine fibroid group should be stated in the Introduction.

2.Materials and Methods

3) A breakdown of the number of cases of each subtype of adenomyosis should be provided.

4) The size of the uterus of the three groups should be listed.

5) The number of cases with heavy menstrual bleeding (menorrhagia) should be listed.

6) The number of myoma knots and subtype (sub serosal, intramuscular, submucosal) should be noted.

7) Page 2, lines 79-81." age, body mass index (BMI), gravidity, parity, menstrual history, signs, symptoms, imaging results, peripheral blood sample collection time, operation type, postoperative pathological report, and so on" It should be listed what should be compared in this study.

8) Page 3, lines 112-114: "2.3. Anemia grade" is listed but not represented in the results. Either these results should be listed or this paragraph should be deleted.

3.Results

9) Table 1: Parity is not listed (if it is not needed, the MM listing should be deleted).

10) Table 1: In the LM group, Dysmenorrhea, Pelvic adhesion is listed as NS, but I don't understand what that means. The number of cases should be stated for each.

11) Table 1: Number of HMB (menorrhagia) is essential (more important than AUB).

12) Table 1: "*" in LM group needs explanation.

13) Table 2 & Fig 2: The values of the results are quite close to each other, were there significant differences? On the other hand, is it meaningful to compare the myoma group with the control group, if there is a meaningful comparison between the three groups, the Kruskal Wallis test, but if only a comparison with the adenomyosis group is needed, it should be done with the Mann-Whitney U test. If there is a meaningful comparison between the three groups, the meaning of that comparison should be stated in the Introduction.

14) Fig 2: "*" in LM group needs explanation.

15) Table 3: "SE" description is missing.

4.Discussion

16) You have not included a discussion of the results obtained from the comparison with uterine myomas. ⇒If that discussion is not necessary, then the comparison is not necessary and all text in this manuscript regarding the comparison with myomas should be deleted.

17) Limitations of this study are not listed. They should be listed.

Abstract

18) The abstract should be revised as the text is revised.

Author Response

Response to Reviewer 1#

This manuscript could be a useful article examining the changes of coagulation parameters and associated factors in women with adenomyosis. However, there is no description of why the changes in coagulation parameters and associated factors in women with uterine adenomyosis should be identified. There is mention in previous papers that they have not yet been clarified. but the manuscript does not indicate the benefit to patients with adenomyosis after it is clarified. It is not important to clarify because it is not known, but what can be done for adenomyosis if it is clarified. A quicker diagnosis? A safe treatment plan?  In other words, it does not describe the medical problems (research questions) caused by the lack of identification of changes in coagulation parameters and associated factors.

Reply: Thanks and we revised it in the introduction section.

It also compares the leiomyoma group as well as the control group, but there is no mention of the need for that comparison. The comparison may have been made because of the similarity to adenomyosis in that it is a benign disease with an enlarged uterus, heavy menstrual bleeding (menorrhagia) and anemia. This may be to indicate whether the cause of the change in coagulation parameters is adenomyosis or heavy menstrual bleeding and/or anemia. However, it is unclear whether the myoma group is worthy of comparison since there is no mention of uterine size, myoma type, or presence of heavy menstrual bleeding. Conversely, the myometrial group in this study had less anemia and significantly higher hemoglobin levels, suggesting that the myometrial group in this study is inappropriate for comparison. Cases of myoma with a uterus approximately the same size as adenomyosis and approximately the same frequency of excessive menstrual periods should be selected (e.g., only intramuscular myomas of the same size as adenomyosis should be selected).

Reply: Thanks and we revised it in the introduction section and discussion section.

  1. Introduction

1)     The lack of clarification of changes in coagulation parameters and related factors in adenomyosis is described by presenting many references, but what would be improved or what would be useful if this were clarified should be described.

Reply: Thanks and we revised it in the introduction section.

2)     The reason for comparing adenomyosis to the control group has been clarified, but the significance of comparing the uterine fibroid group should be stated in the Introduction

Reply: Thanks and we revised it in the introduction section.

2.Materials and Methods

3) A breakdown of the number of cases of each subtype of adenomyosis should be provided.

Reply: Thanks and we only provided focal and diffuse adenomyosis due to our retrospective study, and weakness was explained in the discussion section.

4) The size of the uterus of the three groups should be listed.

Reply: Thanks and we had supplemented the data of uterine size, see Table 1.

5) The number of cases with heavy menstrual bleeding (menorrhagia) should be listed.

Reply: Thanks and we had supplemented the cases of menorrhagia, see Table 1.

6) The number of myoma knots and subtype (sub serosal, intramuscular, submucosal) should be noted.

Reply: Thanks and we only provided the subtype of uterine leiomyoma due to our retrospective study, and weakness was explained in the discussion section.

7) Page 2, lines 79-81." age, body mass index (BMI), gravidity, parity, menstrual history, signs, symptoms, imaging results, peripheral blood sample collection time, operation type, postoperative pathological report, and so on" It should be listed what should be compared in this study.

Reply: Thanks and we revised it.

8) Page 3, lines 112-114: "2.3. Anemia grade" is listed but not represented in the results. Either these results should be listed or this paragraph should be deleted.

Reply: Thanks and we deleted it.

3.Results

9) Table 1: Parity is not listed (if it is not needed, the MM listing should be deleted).

Reply: Thanks and we deleted it.

10) Table 1: In the LM group, Dysmenorrhea, Pelvic adhesion is listed as NS, but I don't understand what that means. The number of cases should be stated for each.

Reply: Sorry for this and we had supplemented the number of cases, see Table 1.

11) Table 1: Number of HMB (menorrhagia) is essential (more important than AUB).

Reply: Thanks and we changed it.

12) Table 1: "*" in LM group needs explanation.

Reply: Sorry for this, and we had supplemented it.

13) Table 2 & Fig 2: The values of the results are quite close to each other, were there significant differences? On the other hand, is it meaningful to compare the myoma group with the control group, if there is a meaningful comparison between the three groups, the Kruskal Wallis test, but if only a comparison with the adenomyosis group is needed, it should be done with the Mann-Whitney U test. If there is a meaningful comparison between the three groups, the meaning of that comparison should be stated in the Introduction.

Reply: Sorry for this, and thanks. We revised it

14) Fig 2: "*" in LM group needs explanation.

Reply: Sorry for this, and we had supplemented it.

15) Table 3: "SE" description is missing.

Reply: Sorry for this, and we had supplemented it.

4.Discussion

16) You have not included a discussion of the results obtained from the comparison with uterine myomas. If that discussion is not necessary, then the comparison is not necessary and all text in this manuscript regarding the comparison with myomas should be deleted.

Reply: Sorry for this, and thanks. We added it in the discussion section.

17) Limitations of this study are not listed. They should be listed.

Reply: Sorry for this, and we had supplemented it.

Abstract

18) The abstract should be revised as the text is revised.

Reply: Thanks and we revised it.

Reviewer 2 Report

I congratulate the authors on this very useful and well-written work. While in recent years several works have addressed the meaning of platelet count (PC), platelet-to-lymphocyte-ratio or other coagulation parameters in endometriosis, less attention has been paid to these phenomena in adenomyosis. Of particular value is the comparison of the parameters studied in patients with uterine fibroids, since the latter are a common pathology that often causes anemia. The Methods section contains all the necessary information. The study design is appropriate, well explained and illustrated with a flowchart. The presentation of the results is logical and comprehensive.  

Some minor suggestions:  

1) Restricting the observations to patients with anemia leads to a transparent study design, but otherwise limits the generalizability of the results. I miss the fact that the authors don't discuss this fact when comparing their own results to other studies. Likewise, the influence of anemia on coagulation parameters should be discussed more thoroughly.  

2) Interestingly, the systematic review by Ottolina (PMID: 32784640, included in the present paper) reported no differences in PC in patients with endometriosis and healthy controls. In addition, another recent study (PMID: 33459084, please specify) reported an even lower mean PC in patients with endometriosis, which is surprising (and deserving futher investigation) but offers another way to use PC (and ancillary measures) to distinguish between adenomyosis, endometriosis, and leiomyoma.

2a) In this context, the conclusion of the work by Coskun et al. is misleadingly simplified here, because the authors suggest that the "case-control study by Coskun et al. found no significant differences in MPV and PC between women << with and without >> adenomyosis". In fact, the more differentiated conclusion of PMID: 31499283 was that PC (and MPV) were not useful diagnostic markers for differentiating between endometriosis or adenomyosis.  

3) And this is where caution is needed: adenomyosis is not the only possible cause of thrombocytosis in patients with gynecologic symptoms. It should always be kept in mind, particularly because an elevated PC may indicate ovarian cancer or other (non-)gynecological malignancies, as shown in PMID: 27207344 or PMID: 26499778. I strongly recommend including this reflection in the discussion to raise awareness of PC (and ancillary parameters) in gynecologic patients.

Author Response

Response to Reviewer 2#

1) Restricting the observations to patients with anemia leads to a transparent study design, but otherwise limits the generalizability of the results. I miss the fact that the authors don't discuss this fact when comparing their own results to other studies. Likewise, the influence of anemia on coagulation parameters should be discussed more thoroughly. 

Reply: Thanks, and we added and revised in the introduction section and discussion section.

2) Interestingly, the systematic review by Ottolina (PMID: 32784640, included in the present paper) reported no differences in PC in patients with endometriosis and healthy controls. In addition, another recent study (PMID: 33459084, please specify) reported an even lower mean PC in patients with endometriosis, which is surprising (and deserving futher investigation) but offers another way to use PC (and ancillary measures) to distinguish between adenomyosis, endometriosis, and leiomyoma.

Reply: Thanks, and we added and revised in the introduction section and discussion section.

2a) In this context, the conclusion of the work by Coskun et al. is misleadingly simplified here, because the authors suggest that the "case-control study by Coskun et al. found no significant differences in MPV and PC between women << with and without >> adenomyosis". In fact, the more differentiated conclusion of PMID: 31499283 was that PC (and MPV) were not useful diagnostic markers for differentiating between endometriosis or adenomyosis. 

Reply: Sorry for this, and we revised it..

3) And this is where caution is needed: adenomyosis is not the only possible cause of thrombocytosis in patients with gynecologic symptoms. It should always be kept in mind, particularly because an elevated PC may indicate ovarian cancer or other (non-)gynecological malignancies, as shown in PMID: 27207344 or PMID: 26499778. I strongly recommend including this reflection in the discussion to raise awareness of PC (and ancillary parameters) in gynecologic patients.

Reply: Thanks, and we added it in the discussion section.

This manuscript is a resubmission of an earlier submission. The following is a list of the peer review reports and author responses from that submission.

Round 1

Reviewer 1 Report

Dear authors. 

Please consider the following comments for revision: 

Introduction:

Line 30: please provide the citation by Leyendecker.

Line 51: please check this phrase.

Methods:

- Please provide information on hormonal treatment of included patients.

- Please specify how you diagnosed adenomyosis or how you excluded adenomyosis in LM group and control group (histology? Ultrasound?MRI?)

- What kind of surgery has been performed? Hysterectomy? Resection of fibroids / AM?

- Please specify how you excluded additional peritoneal and / or deep endometriosis in all patients. You did not mention this detail at all. Does this mean that any of the patients was found to have endometriosis?

Discussion: 

I am not sure if I understand the logic in the first part of the discussion (Line 175 - 194). Please check the first phrase. Please explain the ratio of the citation of Piccioni. Do you want to say that age is an important factor in the severity of adenomyosis related symptoms?

- Please discuss the situation in patients with adenomyosis and additional peritoneal and / or deep endometriosis. What would be the effect?

Author Response

Introduction:

Line 30: please provide the citation by Leyendecker.

Reply: Thanks and we added this reference.

Line 51: please check this phrase.

Reply: Thanks and we revised it.

Methods:

- Please provide information on hormonal treatment of included patients.

- Please specify how you diagnosed adenomyosis or how you excluded adenomyosis in LM group and control group (histology? Ultrasound?MRI?)

- What kind of surgery has been performed? Hysterectomy? Resection of fibroids / AM?

- Please specify how you excluded additional peritoneal and / or deep endometriosis in all patients. You did not mention this detail at all. Does this mean that any of the patients was found to have endometriosis?

Reply: Sorry for this, and we revised and added the study flow diagram (see Figure 1).

Discussion:

I am not sure if I understand the logic in the first part of the discussion (Line 175 - 194). Please check the first phrase. Please explain the ratio of the citation of Piccioni. Do you want to say that age is an important factor in the severity of adenomyosis related symptoms?

Reply: Thanks and we have rewritten the first part of the discussion.

- Please discuss the situation in patients with adenomyosis and additional peritoneal and / or deep endometriosis. What would be the effect?

Reply: Thanks and we added some sentences to describe the effect of endometriosis on the results if adenomyosis coexists with endometriosis. Because we have no endometriosis patients in this study (see Figure 1).

Reviewer 2 Report

Selection of patients with adenomyosis is too unclear.Adenomyosis is diagnosed and confirmed on uterus specimen  and you didn't specify this.

Patients with adenomyosis have often an associated endometriosis. You didn't specify if an associated adenomyosis was an exclusion criteria.

Association between anemia and coagulation is too complex. In this context, it's too hazardous to analyse coagulation variation in the context of anemia without considering multiple factor (cycle phase, iron treatment....).

Author Response

Selection of patients with adenomyosis is too unclear.Adenomyosis is diagnosed and confirmed on uterus specimen  and you didn't specify this.
Patients with adenomyosis have often an associated endometriosis. You didn't specify if an associated adenomyosis was an exclusion criteria.
Reply: Sorry for this, and we revised and added the study flow diagram (see Figure 1).
Association between anemia and coagulation is too complex. In this context, it's too hazardous to analyse coagulation variation in the context of anemia without considering multiple factor (cycle phase, iron treatment....).
Reply: Sorry for this. Unfortunately, our retrospective study can’t carry out a complete multivariate analysis. Therefore,we explained our results in the section of the conclusion and discussion.

Round 2

Reviewer 1 Report

Dear authors, 

thank you for addressing the comments.

I would like to add the following comment: 

Line 206-207: I think it would be better to say: ...may be useful for... (actually this needs to be proven be studies) ...or is a promising therapeutical approach.....

Congratulations for your work. 

Best wishes

Reviewer 2 Report

Dear Author,

This manuscript is too limited concerning different points of methodology such as adenomyosis definition, patient selection. Interpretation is too hazardous as Authors considered anemia and coagulation variations as independants.